# Slow growing behavior in African trypanosomes during adipose tissue colonization

Sandra Trindade[1], Mariana De Niz[1], Mariana Costa-Sequeira [1], Tiago Bizarra-Rebelo[1], Fábio Bento[1,6], Mario Dejung[2], Marta Valido Narciso [1], Lara López-Escobar [1], João Ferreira[1], Falk Butter [2], Frédéric Bringaud[3,4], Erida Gjini[5,7] ✉ & Luisa M. Figueiredo [1] ✉

When *Trypanosoma brucei* parasites, the causative agent of sleeping sickness, colonize the adipose tissue, they rewire gene expression. Whether this adaptation affects population behavior and disease treatment remained unknown. By using a mathematical model, we estimate that the population of adipose tissue forms (ATFs) proliferates slower than blood parasites. Analysis of the ATFs proteome, measurement of protein synthesis and proliferation rates confirm that the ATFs divide on average every 12 h, instead of 6 h in the blood. Importantly, the population of ATFs is heterogeneous with parasites doubling times ranging between 5 h and 35 h. Slow-proliferating parasites remain capable of reverting to the fast proliferation profile in blood conditions. Intravital imaging shows that ATFs are refractory to drug treatment. We propose that in adipose tissue, a subpopulation of *T. brucei* parasites acquire a slow growing behavior, which contributes to disease chronicity and treatment failure.

*Trypanosoma brucei* parasites, the causative agents of Human African Trypanosomiasis (HAT), have a complex life cycle that oscillates between a tsetse vector and a mammalian host. The quiescent non-dividing stumpy forms and metacyclic forms ensure transmission between them. During the mammal infection, parasites invade and occupy the interstitial spaces of different organs, such as the central nervous system, adipose tissue and skin[1,2]. The adaptation to the tissue environment promotes heterogeneity within a *T. brucei* population. We have previously shown that, in a mouse infection, adipose tissue is one of the largest reservoirs of *T. brucei* parasites[3,4] occupied by a population of parasites that we named adipose tissue forms (ATFs). These forms are functionally different from their bloodstream counterparts (BSFs): ATFs have a metabolism apparently adapted to catabolize fatty acids and are capable of re-infecting the blood circulation, showing that they are not terminally differentiated to live exclusively in the adipose tissue[3].

Many microorganisms have evolved mechanisms that ensure their survival in different environmental conditions. The best studied mechanism is persistence, which was first described in bacteria. Persisters consist of a subgroup of viable cells within a population that can arise stochastically or in response to environmental cues. They are typically characterized by growth arrest or slower growth, altered metabolism, reduced protein synthesis and increased resistance to environmental threats such as immunological attack and drug

[1]Instituto de Medicina Molecular João Lobo Antunes, Faculdade de Medicina, Universidade de Lisboa, 1649-028 Lisbon, Portugal. [2]Institute of Molecular Biology, 55128 Mainz, Germany. [3]Laboratoire de Microbiologie Fondamentale et Pathogénicité (MFP), Université de Bordeaux, CNRS, UMR-5234 Bordeaux, France. [4]Centre de Résonance Magnétique des Systèmes Biologiques (RMSB), Université de Bordeaux, CNRS, UMR-5536 Bordeaux, France. [5]Instituto Gulbenkian de Ciência, 2780-156 Oeiras, Portugal. [6]Present address: Institute of Molecular Biology, Mainz, Germany. [7]Present address: Center for Computational and Stochastic Mathematics, Instituto Superior Técnico, Universidade de Lisboa, 1049-001 Lisbon, Portugal. ✉e-mail: erida.gjini@tecnico.ulisboa.pt; lmf@medicina.ulisboa.pt

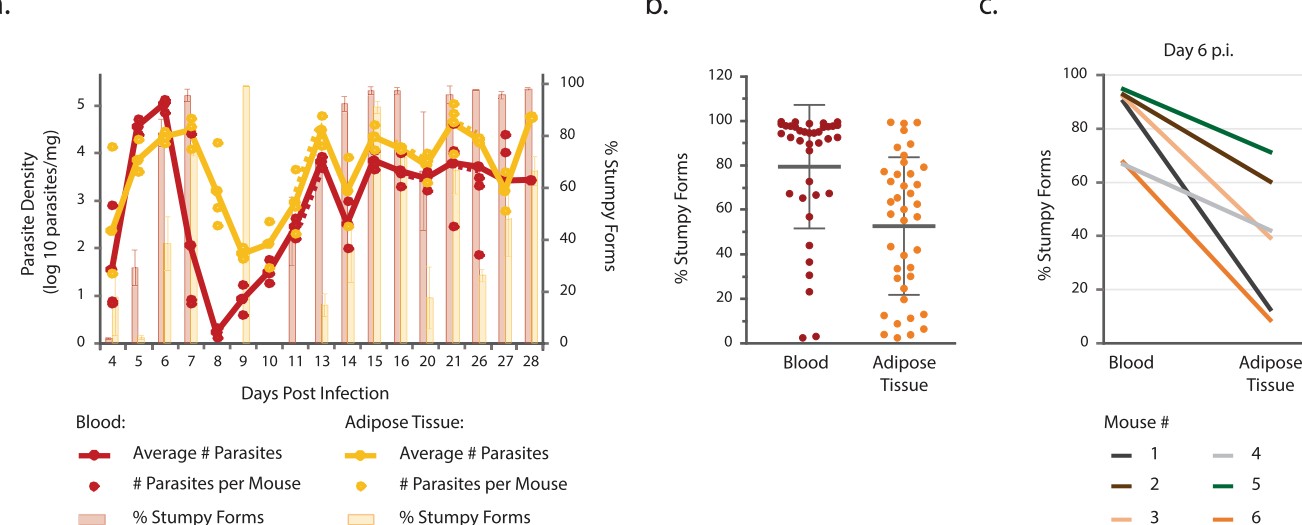

**Fig. 1 | Infection dynamics in blood and gonadal adipose tissue. a** C57BL/6J mice were infected with a pleomorphic *T. brucei GFP::PAD1_{utr}* stumpy reporter cell-line. The number of parasites per milligram of tissue was quantified by qPCR (red and yellow lines and dots) and the percentage of stumpy forms was estimated by flow cytometry as the proportion of GFP-positive parasites (red and yellow bars). Parasite density: $n = 3$ independent experiments for all time points, except for day 21 in which $n = 4$ independent experiments. Percentage of stumpy forms: $n = 3$ independent experiments for all time points, except for day 6 in which $n = 7$ independent experiments and day 7 in which $n = 5$ independent experiments. The dashed lines correspond to noncontiguous time points and the full lines to contiguous time points. Error bars represent the standard error of the mean. Occasionally, the parasite density was too low which resulted in $n = 2$. **b** Percentage of stumpy forms in blood (red circles) and gonadal adipose tissue (yellow circles) for the 17 studied time points ($n = 38$ BSFs and $n = 40$ ATFs pooled from all independent experiments from all time points). Gray lines represent the average percentage of stumpy forms and error bars the standard deviation, two-sided Wilcoxon-signed rank test, $p = 1.205e-8$. **c** Percentage of stumpy forms in blood and gonadal adipose tissue of six mice sacrificed at day 6 of infection. Source Data are provided as a Source Data file.

treatment. This condition is reversible and persisters can return to a more proliferative state[5,6]. Among parasitic protozoa, the existence of persisters has only been recently documented, and knowledge on the biological roles of these parasite subpopulations is still scarce[7]. *Toxoplasma gondii*, a protozoan parasite that infects most species of warm-blooded animals, upon host invasion, disseminate into tissues and differentiate into persistent metabolically active, slow growing bradyzoites as a strategy to overcome nutrient limitations and host immune pressure[8]. Persistence can also be triggered by temporary exposure to other external stresses. *P. falciparum*, for instance, develops into dormant ring stages with a distinct metabolic state upon artemisinin exposure[9]. Importantly, these persistent parasitic protozoa are refractory to artemisinin-based drugs promoting drug treatment failure[7]. Phenotypic plasticity is another mechanism employed to surpass environmental disturbances presented by changing environments. Plastic organisms have the ability to transiently reduce growth as an adaptation process. Contrary to the typical heterogeneity observed in persister populations, plastic populations are homogeneous, changing as a whole with an environmental shift[10,11].

In this work, we use mathematical modelling to determine key parameters that govern *T. brucei* chronic infections taking into account the existence of tissue reservoirs. We experimentally confirmed the prediction of the model for differences between reservoirs, showing that the adipose tissue harbors a population that synthesizes proteins at a lower rate and grows heterogeneously with a significant large subgroup of slow growing parasites. Importantly, adipose tissue forms are able to survive drug treatment, while parasitemia becomes undetectable. Together with previous knowledge that ATFs are metabolically distinct from their blood counterparts[3], we propose the existence of slow growing parasites within the adipose tissue population. This finding has important implications to our understanding of drug resistance and relapses, and reveals that adipose tissue acts as a reservoir of quieter parasites that may cause less pathology, contributing to disease chronicity.

## Results

### Parasite dynamics differ between blood and adipose tissue

We previously observed that, on day 6 of infection, the adipose tissue is mainly populated by slender forms, while the blood has a majority of transmissible stumpy forms[3]. Whether the adipose tissue stands as major reservoir and the proportion of slender and stumpy forms varies in the two tissues, throughout infection, remains unknown. To answer this question, we infected mice with a pleomorphic *GFP::PAD1_{utr}* reporter parasite line, which allowed us to follow in each tissue and for over 28 days of infection not only the parasite density, but also the proportion of parasites that have initiated the process of differentiation to transmissible stumpy forms (intermediate and stumpy forms are GFP-expressers)[3]. Mice were sacrificed at multiple days of infection and the number of parasites per milligram of organ (parasite density) was quantified by qPCR. BSFs and ATFs were also isolated from both tissues to assess the percentage of stumpy forms by flow cytometry.

In general, both tissues remained highly parasitized throughout infection. The parasite density in the gonadal adipose tissue was on average ~6-fold higher than in the blood (Supplementary Table 1). As previously observed[3,12,13], during the first week of infection, the blood showed a larger range of parasite density that peaked at day 6 (~$10^5$ parasites/mg) and a subsequent trough on day 8 (~2 parasites/mg). These results showed an initial slower colonization of the adipose tissue and a less pronounced reduction of this population after day 6 of infection. From day 13 forward, the parasite densities were less variable and the pattern more similar between tissues (~$10^3$–$10^4$ parasites/mg), with the adipose tissue having on average a 4-fold higher parasite density than the blood (Fig. 1a and Supplementary Table 1).

Next, we assessed if the proportion of replicative (slender) and non-replicative transmissible (stumpy) forms is different between the two tissues throughout infection. Parasites were isolated from blood and gonadal adipose tissue and flow cytometry was used to quantify the proportion of GFP-expressing parasites[14]. Except for day 4, the blood was always richer in stumpy forms than the gonadal adipose

a.

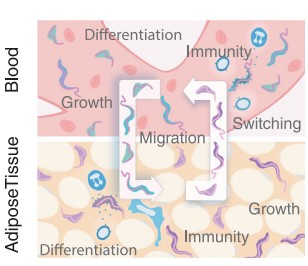

b.

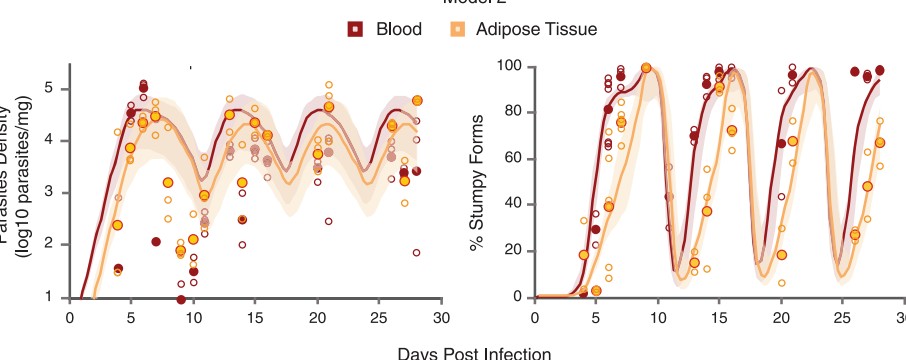

**Fig. 2 | Mathematical model for parasite dynamics in blood and gonadal adipose tissue. a** Diagram of the mathematical infection model. In this model, two compartments were considered: blood and adipose tissue. In each compartment, parasites may have intrinsic growth and differentiation rates. A constant migration rate was also considered between the two. The specific immune response was assumed to be triggered due to overall antigen stimulation from blood and adipose tissue parasites, after which it grows in response to increasing pathogen density and reaches each compartment (Materials and Methods and Supplementary Model Information). **b** Model fit for predictions of parasite density and proportion of stumpy forms, in blood and adipose tissue, for the favored model (model 2). Scattered empty circles indicate all observations from individual time-points and filled circles indicate the mean over replicates used to fit the model. The lines indicate the model prediction with mean parameters (as in Table 1). The shaded regions indicate the 95% credible envelopes from 50 simulations with random parameter combinations from the estimated posteriors. Source Data are provided as a Source Data file.

tissue. In general, in the adipose tissue, the average percentage of stumpy forms was lower (52% vs 79%; Wilcoxon-signed rank test, $p = 1.205e{-}8$) and more evenly distributed than in the blood, where a population with ~100% stumpy forms is more prevalent (Fig. 1b). Interestingly, for the first two weeks of infection, the accumulation of stumpy forms in the adipose tissue proceeds more slowly than in the blood. Data from individual mice showed that the proportion of stumpy forms can be dramatically different between the two micro-environments (Fig. 1c), indicating that within a single animal there are significant differences in the proportion of stumpy forms across tissues.

We conclude that the dynamics of the number of parasites and the proportion of slender/stumpy forms are different between blood and gonadal adipose tissue. These results suggest an influence from the tissue microenvironment and the existence of an important barrier (likely, the vasculature) that prevents homogenization of the two parasite populations.

### Infection modeling anticipates differences in parasite growth between tissues

To quantify the infection processes in more detail, and investigate which factors may contribute to differences between microenvironments, we built a mathematical model to integrate under the same mechanistic framework the empirical data on the parasite density and the proportion of stumpy forms across the two compartments over time (Fig. 2a, Supplementary Table 1, Supplementary Data 1, and Supplementary Model Information). Some model parameters were assumed fixed at biologically reasonable values (Supplementary Table 2). Three nested versions of the model were considered. In model 1 (the null hypothesis), no infection parameter differences between compartments were assumed (except for the blood exclusive onset of infection and antigen switching); in model 2, the parasites may grow at different rates in each compartment; in model 3, in addition to growth rates, differentiation parameters could also vary across the two tissues. Although the model structure is flexible enough to explore many more hypotheses about parameter differences between compartments, we limited our analysis to these three basic formulations, differing sequentially in just one parameter, which are the simpler and easier to validate with additional experiments.

We fitted each model dynamically to the experimental data under a Bayesian framework, and then compared their results based on the Deviance Information Criterion (DIC)[15], the likelihood ratio[16], as well as the visual inspection of the 95% credible envelopes for the infection trajectories. Model fits to experimental data generated model-specific estimates for infection parameters and corresponding predictions for the inter-coupled infection dynamics between blood and adipose tissue, even though the models are realistically close to each other (Table 1 and Supplementary Model Information). The DIC criterion and the likelihood ratio test were the statistical indicators that allowed comparing how well the models fit the raw data, balancing quality of fit with model complexity. Typically, models with a smaller DIC are favored. In addition, a large likelihood ratio would favor the more complex model while a small likelihood ratio would favor the simpler model. For this specific dataset, model 2 was favored over models 1 and 3 (Supplementary Table 3) by both the above criteria. The model was able to capture accurately the global parasite dynamics across the blood and adipose tissue over 28 days (Fig. 2b and Supplementary Model Information). Further adding an extra layer of complexity, by allowing variation of the differentiation parameter across the two compartments (model 3), produced largely overlapping estimates for these infection traits in blood and adipose tissue (Table 1) and did not significantly improve the model fit (Supplementary Table 3).

The favored model 2 estimated a doubling time of 7 h and 13.7 h for parasites residing in the blood and adipose tissue, respectively. This observation is in good agreement with previous modeling where bloodstream form parasites were estimated to divide every 6 h[17]. The inferred 50% lower growth rate of slender forms in the adipose tissue (Table 1) enabled a better description of the observed slow buildup of stumpy forms over time in this tissue. This indicates that allowing for heterogeneous replication of slender form parasites between the blood and adipose tissue is both necessary and sufficient to capture the experimental global patterns (Fig. 2b and Supplementary Model Information).

Overall, we were able to integrate in a dynamic model the experimental data on *T. brucei* parasite density and the proportion of stumpy forms in blood and adipose tissue of infected mice. The best-fitting model indicates that the parasites residing in the adipose tissue replicate at half the rate of those in the blood.

**Table 1 | Mathematical models parameters and estimates**

| Parameter | Interpretation | Estimates[a] | | | Units |
|---|---|---|---|---|---|
| | | Model 1 | Model 2 | Model 3 | |
| $r$ | Growth rate | 2.12 (7.8[b]) | – | – | Divisions/per day |
| | | (1.85; 2.40) | – | – | – |
| $R_b$ | Growth rate of slender forms in blood | – | 2.38 (7.0[b]) | 2.33 (7.1[b]) | Divisions/per day |
| | | – | (2.10; 2.56) | (2.14; 2.49) | – |
| $R_f$ | Growth rate of slender forms in adipose tissue | – | 1.21 (13.7[b]) | 1.23 (13.5[b]) | Divisions/per day |
| | | – | (1.01; 1.56) | (1.01; 1.69) | – |
| $d$ | Minimal killing rates for slender forms[c] | 0.0016 | 0.0032 | 0.0029 | Cells/mg/day |
| | | (0.0001; 0.0063) | (0.0005; 0.0066) | (0.0006; 0.0065) | – |
| $d'$ | Minimal killing rates for stumpy forms[c] | 0.0005 | 0.0001 | 0.0002 | Cells/mg/day |
| | | (0.0001; 0.0028) | (0.00005; 0.0008) | (0.00005; 0.0009) | – |
| $K$ | Density for maximal differentiation | 12,600 | 31,300 | – | Cells/mg |
| | | (3400; 51,000) | (6700; 58,700) | – | – |
| $K_b$ | Density for maximal differentiation in blood | – | – | 26,400 | Cells/mg |
| | | – | – | (4000; 58,100) | – |
| $K_f$ | Density for maximal differentiation in adipose tissue | – | – | 14,200 | Cells/mg |
| | | – | – | (3300; 54,200) | – |
| $k$ | Density for half-saturation immune stimulation | 830 | 3600 | 3400 | Cells/mg |
| | | (210; 6440) | (990; 16,600) | (1200; 14,000) | – |
| $\sigma$ | Activation rate of anti VSG immune response | 1.38 | 1.68 | 1.67 | Per day |
| | | (1.04; 1.83) | (1.27; 2.21) | (1.28; 2.21) | – |
| $\mu$ | Migration rate across compartments | 0.0117 | 0.1075 | 0.0633 | Cells/mg/day |
| | | (0.0069; 0.0311) | (0.0523; 0.1901) | (0.0101; 0.8353) | – |
| $s$ | Switch probability to next antigenic wave in blood | 0.0007 | 0.0004 | 0.0006 | Per division |
| | | (0.0001; 0.0051) | (0.0001; 0.0014) | (0.0002; 0.0019) | – |

[a]Estimates are given as means and 95% Confidence Intervals are indicated in italic.
[b]Doubling time (hours).
[c]Number of killed cells by 1 unit of immune response due to the feedback from VSG-specific immunity.

### Adipose tissue forms have reduced protein synthesis

To experimentally validate the model prediction that slender forms in the adipose tissue replicate at half the rate of parasites in the blood, we compared the proteome of slender forms isolated from blood and gonadal adipose tissue. Given that the proteome of stumpy and slender forms is different[18], and stumpy forms are present in unequal amounts in the two tissues, we chose to infect mice with a monomorphic strain, which does not form stumpy forms. Five days postinfection, parasites were isolated from blood and adipose tissue and processed for proteome analysis.

We identified a total of 2693 protein groups (PGs) among ATFs and BSFs (Supplementary Data 2), from which 6% were differentially expressed, namely, 54 PGs and 112 PGs were up- and downregulated, respectively, in the ATFs (Fig. 3a). The relative abundances of the differentially expressed PGs showed high consistency between replicates (Fig. 3b), suggesting that the differences detected between ATFs and BSFs are reproducible between infections in different animals.

GO term enrichment analysis of the proteins upregulated in ATFs suggests that these parasites are metabolically distinct from BSFs (Fig. 3c and Supplementary Table 4), which is consistent with our previous transcriptomic and biochemical characterization[3]. Interestingly, the most significant category of upregulated proteins comprised purine metabolism (5/54 upregulated PGs; Fig. 3c, Supplementary Data 2, and Supplementary Table 4) which may reflect a need to increase the ATP production capacity in the adipose tissue environment. Interestingly, ATFs seem to share some features with stumpy forms, while they also have important differences. Like stumpy forms, ATFs appear to show an increased production of ATP, via the acetate

pathway[19], and an upregulation of the enzymes of the glutamine/proline degradation pathway[20,21]. Unlike stumpy forms, ATFs did not show evidence for upregulation of the Protein Associated with Differentiation 1 (PAD1), Expression Site Associated Gene 9 (ESAG9), Protein Phosphatase 1 (PP1), or RNA-binding Protein 7 (RBP7)[22]. Mitochondrial acetate production and glutamine catabolic pathways, both involved in ATP metabolism, are also upregulated (9/54 upregulated PGs; Fig. 3c, Supplementary Data 2, and Supplementary Table 4). Reversely, an increased mitochondrial ATP consumption may also occur given the lack of detection of any component of the oxidative phosphorylation pathway and the expanded use of the mitochondrial $F_1/F_O$-ATP synthase (4/54 upregulated PGs; Fig. 3c, Supplementary Data 2, and Supplementary Table 4).

Among the proteins downregulated in ATFs, translation and ribosome biogenesis (pivotal for protein translation) were the two significant GO terms identified (Fig. 3c and Supplementary Table 4). 96 ribosomal proteins of 208 annotated protein groups were downregulated in the ATFs proteome (Supplementary Data 2). A curated analysis of the proteins revealed that multiple ribosomal proteins of both 40S and 60S ribosomal subunits were downregulated. Interestingly, despite not annotated to any of the obtained GO terms, the second largest subunit of the RNA polymerase I was also downregulated (Tb927.11.630, fold change of 0.5) suggesting a lower transcription of ribosomal gene units (Supplementary Data 2 and Supplementary Table 4). These data strongly suggest that ATFs undergo a generalized reduction in the machinery responsible for synthesizing proteins.

To investigate whether ATFs really have a lower rate of protein synthesis we measured protein synthesis by labeling newly synthesized

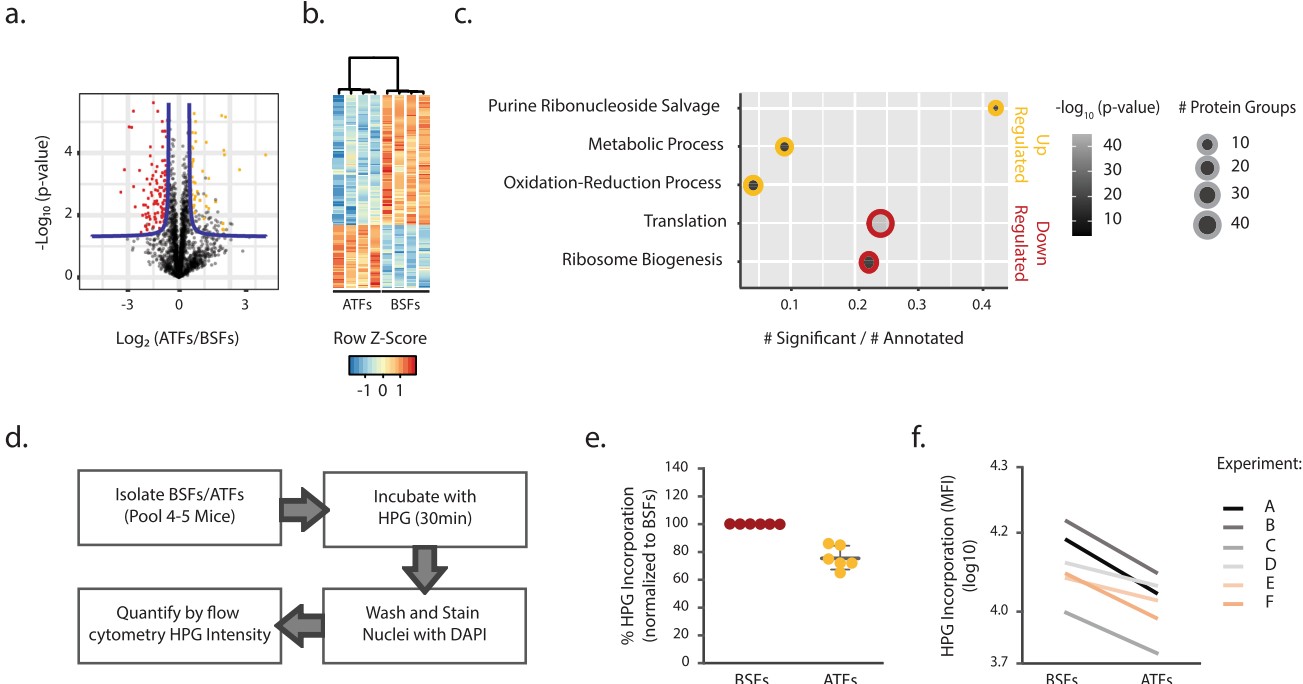

**Fig. 3 | Proteome analysis and protein synthesis of parasites isolated from blood and gonadal adipose tissue. a** Representative Volcano plot of 1 out of the 1000 performed imputations with the 2693 protein groups detected by mass spectrometry in the sum of BSFs and ATFs 8 data sets. Red and yellow dots indicate protein groups significantly upregulated in BSFs and ATFs, respectively. Proteins were obtained from Lister 427 parasites isolated from a pool of 5–6 animals infected for 5 days ($n = 4$ independent experiments). **b** Heat map representing the z-score abundances of the differentially expressed protein groups. Each column corresponds to one independent pool of mice. **c** Enriched GO terms (Biological Process) of the differentially upregulated (yellow circles) and downregulated (red circles) proteins according to gene ratio, using a False Discovery Rate corrected p-value cut-off of 0.05 (two-sided Fisher's exact test). Each dot corresponds to a given GO term: the size represents the number of differentially expressed protein groups and the color the respective p-value. **d** Fluxogram of the experimental procedure used to quantify protein synthesis in Lister 427 parasites isolated from blood and gonadal adipose tissue 5 days post-infection. **e** Percentage of HPG internalized by ATFs (yellow circles) normalized to the percentage of internalization of the BSFs (red circles) ($n = 6$ independent experiments). Gray lines represent the average percentage of HPG incorporation (normalized to BSFs) and the error bars represent the standard deviation. **f** Mean Fluorescence Intensity of the HPG incorporated for 30 min by the isolated BSFs and ATFs for experiments A, B, C, D, E, and F ($n = 6$ independent experiments). Two-sided Wilcoxon-signed rank test, $p = 0.03125$. Source Data are provided as a Source Data file.

proteins with a methionine analog, Click-iT® Homopropargylglycine (HPG), in parasites isolated from each of the two tissues. Levels of labeled nascent proteins in BSFs and ATFs were measured, through the Mean Fluorescence Intensity (MFI) values, by flow cytometry. First, we optimized this method for *T. brucei* by testing the linearity of the assay, comparing the protein synthesis between BSFs and Procyclic Forms (PCFs) and inhibiting protein synthesis of cultured parasites with cycloheximide. A dose-dependent response in which the levels of labeled proteins increased with the levels of HPG (Supplementary Fig. 1A) allowed us to define the use of 50 μM of HPG in all subsequent experiments. The HPG-labeling assay revealed that PCFs incorporated on average 74% less HPG than BSFs (Supplementary Fig. 1B), (following the reported trend[23]), and that cycloheximide promoted a drop of 93% on HPG intensity (Supplementary Fig. 1C). These observations validate the assay by showing that less protein synthesis result in less HPG incorporation and excluding the possible interference of free HPG inside the cell.

To compare the protein synthesis of parasites from different tissues, infected mice were sacrificed on day 5 post-infection and parasites were isolated from blood and gonadal adipose tissue in parallel. In each of the six independent experiments, parasites from 4–5 mice were pooled prior to HPG labeling. After incubation for 30 min with HPG, parasites were washed and analyzed as indicated in Fig. 3d. Flow cytometry revealed that the intensity of HPG signal was 24% lower in parasites isolated from adipose tissue than from blood (Fig. 3e). In fact, the intensity of HPG-labeled proteins was always lower in ATFs than BSFs (with a range of reduction between 14 and 35%; Fig. 3f; Wilcoxon-

signed rank test, $p = 0.03125$), indicating that ATFs have a reduced protein synthesis.

In summary, the slender forms that live in the blood and adipose tissue present significantly different proteomes and the latter population shows a reduced rate of protein synthesis.

## Adipose tissue forms proliferate heterogeneously

In many eukaryotes, a reduced rate of protein synthesis is normally associated to slower growth[24,25]. Our modeling analysis indeed suggested that, on average, unlike BSFs (7 h), ATFs replicate every 13 h 42 min (Table 1). To test whether ATFs replicate more slowly than BSFs, we started by analyzing their cell cycle patterns in vivo.

First, we compared the number of kinetoplasts/nuclei (KsNs)[26] in BSFs and ATFs. Intravenous administration of Hoechst to mice infected for 5 days with a monomorphic strain followed by intravital imaging (Fig. 4a and Supplementary Movie 1) revealed that ATFs have on average a significantly higher percentage of parasites presenting 1K1N (cells in G1, early S phase or G0-arrested cells) (73.6% vs 60.1% in BSFs) and 2K1N (G2 and mitotic cells; 17.1% vs 11.6% in BSFs) configurations. The most significant difference was that ATFs have a lower percentage of 2K2N parasites (post-mitotic; 9.3% vs 28.3% in BSFs; Fig. 4b). These observations suggest that in a snapshot of the ATFs population it is less likely to find parasites that just underwent cell division than in a BSFs population, which would be expected if ATFs proliferate more slowly. Labeling and imaging of isolated parasites ex vivo, showed similar results, indicating consistency between ex vivo and in vivo analysis (Supplementary Fig. 2A).

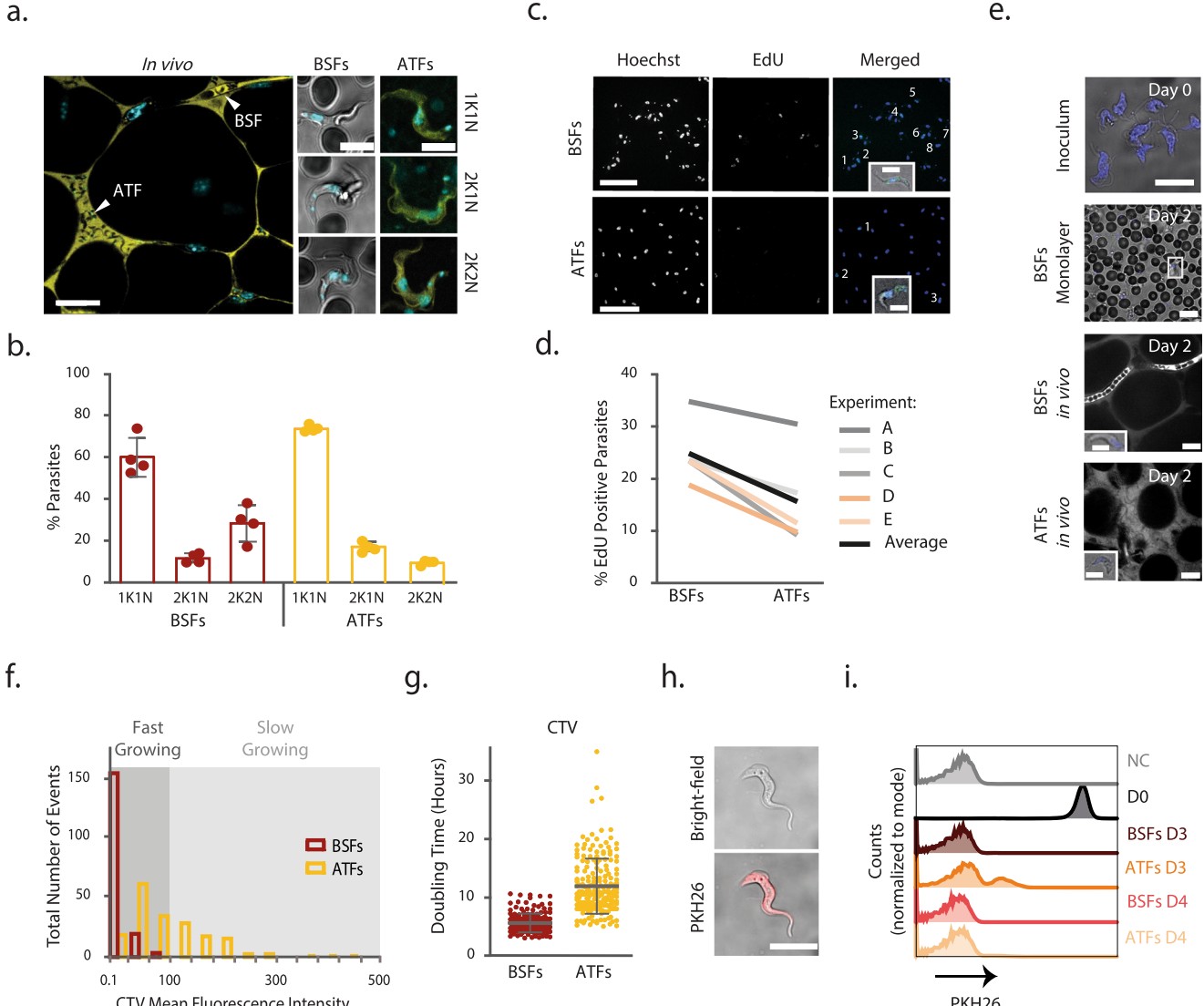

**Fig. 4 | Parasite proliferation in blood and gonadal adipose tissue. a** Cell-cycle analysis assayed by the number of Kinetoplasts (K) and Nuclei (N) in BSFs and ATFs Lister 427 parasites from mice infected for 5 days (n = 4 independent experiments). Left side: Representative image of in vivo microscopy of the adipose tissue and a vessel (scale bar, 20 μm); Right side: representative images of the three parasites analyzed on a monolayer (BSFs) and in vivo (ATFs) (Hoechst, blue; scale bar, 5 μm). **b** Percentage of BSFs (red circles, bars) and ATFs (yellow circles, bars) in each stage of the cell cycle. More than 800 cells per condition. Error bars represent the average and the standard deviation. **c** Ex vivo microscopy images of Lister 427 BSFs and ATFs labeled with EdU (green) and Hoechst (blue) (n = 5 independent experiments) (scale bar, 20 μm). Insets: representative images of labeled parasites (scale bar, 5 μm). **d** Percentage of EdU positive BSFs and ATFs. More than 300 cells per condition. Generalized linear mixed-effects model [GLMER], p = 0.00031. **e** Representative images of ex vivo microscopy of the inoculum parasites and a monolayer of BSFs stained with CTV and in vivo microscopy of ATFs and BSFs (scale

bar, 20 μm). Insets: Representative images of labeled parasites (scale bar, 5 μm). **f** Distribution of the CTV Mean Fluorescence Intensity of BSFs and ATFs from mice infected for 2 days assessed by intravital microscopy. First bin of 20, remaining bins of 40 (n = 183 BSFs and n = 196 ATFs pooled from the 4 independent experiments). **g** Doubling time estimated by in vivo microscopy analysis of Lister 427 BSFs (red circles) and ATFs (yellow circles).(n = 183 BSFs and n = 196 ATFs pooled from the 4 independent experiments). Two-sided linear mixed-effects model [LME], p < 0.0001. Gray lines represent the average values and error bars the standard deviation. **h** Representative image of ex vivo microscopy of cultured Lister 427 parasites labeled with PKH26 (red) (scale bar, 10 μm) (n = 2 independent experiments). **i**, Representative flow cytometry profiles of non-labeled and PKH26 labeled cultured Lister 427 parasites and BSFs and ATFs isolated from mice infected for 3 and 4 days (n = 3 independent experiments) NC: Negative Control, D0: Day0, D3: Day3, D4: Day 4. Source Data are provided as a Source Data file.

To further confirm the differences of cell cycle between BSFs and ATFs, we isolated parasites from both tissues and measured the proportion of the parasites actively synthetizing DNA. For that, mice infected for 5 days with the same monomorphic strain were intravenously injected with 5-ethynyl-2-deoxyuridine (EdU), a thymidine analog that is incorporated into nascent DNA[27] and isolated BSFs and ATFs were analyzed by microscopy for the percentage of EdU+ cells (Fig. 4c). The ATFs showed on average a lower percentage of labeled cells (15.8% vs 24.6%) (generalized linear mixed-effects model [GLMER], p = 0.00031; Fig. 4d), indicating that this

population has fewer cells synthetizing DNA. Ex vivo labeling of isolated parasites corroborated our in vivo observations, showing that isolation does not affect DNA synthesis (Supplementary Fig. 2B).

Next, we compared the proliferation profile of BSFs and ATFs at single-cell level using CellTrace™ Violet (CTV), a fluorescent dye that binds to free amines inside cells (Fig. 4e). Every time a cell divides, the amount of CTV is halved[28]. To test if CTV has any detrimental effect on parasite fitness, we scored parasitemia, by microscopy, and parasite density and motility in the adipose tissue by intravital imaging. CTV

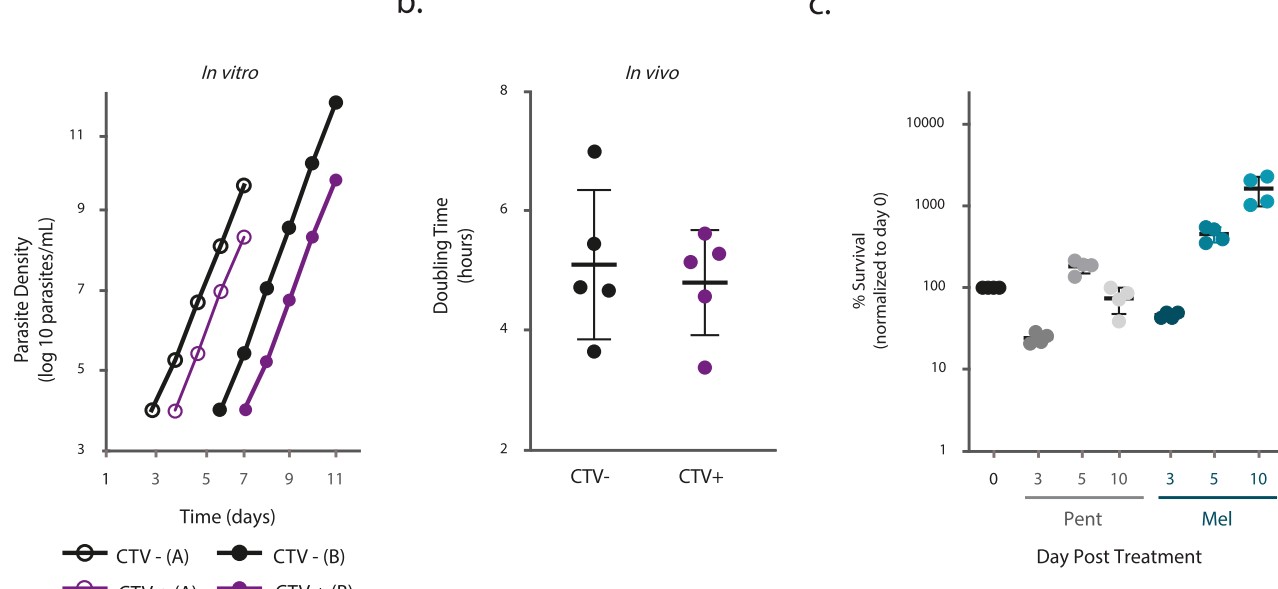

**a.** **b.** **c.**

**Fig. 5 | Characterization of slow growing ATFs. a** In vitro growth curves of sorted CTV negative (CTV−; black) and positive (CTV +; violet) parasites isolated from adipose tissue of mice infected for 3 days with cultured Lister 427 parasites labeled with CTV immediately prior to infection (*n* = 2 independent experiments). **b** In vivo doubling times of sorted CTV negative (CTV−; black) and positive (CTV+; violet) parasites isolated from adipose tissue of mice infected for 3 days with cultured Lister 427 parasites labeled with CTV immediately prior to infection (*n* = 5 independent experiments). Black lines represent the average values and error bars the

standard deviation, (two-sided Wilcoxon rank sum test, *p* = 0.8413). **c** Percentage of survival of adipose tissue parasites assessed by intravital microscopy at 0 (black dots) and 3, 5, and 10 days post mice treatment with 20 mg/kg of pentamidine (gray dots) and 10 mg/kg of melarsoprol (blue dots) administered intraperitoneally at day four post infection with Lister 427 parasites (*n* = 4 independent experiments). Black lines represent the average values and error bars the standard deviation. Source Data are provided as a Source Data file.

labeling did not promote any significant changes in parasite density and motility (Supplementary Fig. 2C) indicating that CTV is not toxic. Importantly, the proliferation profile showed that both populations are made up of parasites that have divided (Supplementary Fig. 2D). A more detailed analysis revealed that the ATFs population is more heterogeneous with MFIs oscillating between 3.7 and 436.8, while the BSFs MFIs oscillated only between 0.1 and 72.6. Indeed, on average 40% of the studied ATFs population displayed exclusive high intensities of CTV (Fig. 4f), indicating that a significant fraction of the ATFs has a slower replication pattern than BSFs. Within ATFs, some parasites divide as fast as the blood parasites, while others divide much slower, with some having up to 35 h of doubling time. On average, the doubling time of the ATFs population was significantly higher (12 h 6 min vs 5 h 42 min) (linear mixed-effects model [LME], *p* < 0.0001; Fig. 4g). Importantly, the calculated doubling time of BSFs was in good agreement with previous reports for both in vivo and in vitro growth[29,30]. An ex vivo analysis, corroborated the intravital microscopy observations (10 h 48 min for ATFs vs 6 h 12 min for BSFs), excluding a possible interference of the staining background detected in vivo (Supplementary Fig. 2E).

To test if the population of ATFs contained a subpopulation of non-dividing cells, we used a lipophilic membrane dye (PKH26) which, in theory, allows tracing proliferation for less cell divisions than CTV. First, we determined if the tracer has the ability to track cell divisions in *T. brucei* cultured Lister 427 parasites labeled with 1 μM of the dye. Microscopy images acquired immediately after labeling confirmed homogeneous parasite labeling (Fig. 4h). Flow cytometry analysis for up to 50 h revealed the expected decreasing pattern of PKH26 MFIs along with accumulating cell divisions (Supplementary Fig. 2F) and an average expected doubling time of 4 h 48 min, validating the method in *T. brucei*. Next, we analyzed by flow cytometry the dye intensity in isolated BSFs and ATFs 3 and 4 days post-infection. While BSFs showed no PKH26 staining, suggesting they have undergone more than seven cell divisions, ATFs showed a delayed pattern: an initial subset of the

population with detectable levels that disappeared the next day (Fig. 4i and Supplementary Fig. 2G). These results confirmed that the adipose tissue harbors a subpopulation of parasites that proliferate more slowly than BSFs. We did not detect a subpopulation of cell cycle arrested ATFs.

Overall, these results show that the population of parasites that colonize the adipose tissue proliferate, on average, at half the rate of the bloodstream form population. Importantly, the proliferation profile within the adipose tissue population is surprisingly heterogeneous, with some parasites proliferating as fast as bloodstream forms and others taking up to 35 h to replicate.

### Adipose tissue forms partially evade drug treatment

We have previously shown that parasites isolated from adipose tissue (and other organs) can establish an infection in a new mouse[3], suggesting that at least some tissue parasites can readapt to bloodstream conditions. Whether fast, slow replicating or all ATFs are capable of adapting to blood conditions remains unknown. For that, we infected mice with CTV-labeled parasites, and 3 days later, we isolated ATFs, we FACS-sorted CTV-positive (slow-replicating parasites) and CTV-negative (fast-replicating parasites) ATFs and either place them in culture or infected mice to follow parasite growth. In vitro, the very small number of FACS-sorted parasites and a putative period of adaptation to the new growth conditions result in an unquantifiable number of parasites for a variable number of days. After this period, the in vitro analysis showed that the growth rate of CTV-negative and CTV-positive parasites was similar (average doubling time of 6 h vs 6 h 30 min; Fig. 5a), indicating that both populations can adapt to the new conditions and grow at the same rate. In vivo, after a period of undetectable parasitemia, both subpopulations displayed a similar growth rate (5 h 6 min ± 1 h 15 min and 4 h 48 min ± 54 min for CTV-negative and CTV-positive, respectively) (Wilcoxon Rank Sum test, *p* = 0.8413; Fig. 5b). Although we cannot exclude the possibility of a

bloodstream-contaminating parasite, these results support the hypothesis that both fast and slow-proliferating ATFs are capable of adapting to blood conditions and thus none are irreversibly committed to living in the adipose tissue.

If a subpopulation of ATFs grows more slowly, it could be more resistant to drug treatment and contribute to treatment failure. To test if ATFs are more resistant to drug treatment than BSFs, mice infected for 4 days with Lister 427 parasites (and with an average parasitemia of 23 million parasites/mL of blood) were treated with non-curative doses (known to only clean the blood infection) of two chosen drugs. A single intraperitoneal injection of 20 mg/kg of pentamidine or 10 mg/kg of melarsoprol was administered. Parasitemia was evaluated for the following 3 days by microscopy and parasite load in the extravascular spaces of the adipose tissue determined by intravital microscopy for 3, 5, and 10 days post treatment (Fig. 5c). The parasitemia dropped to undetectable levels up to 3 days post treatment. In animals treated with pentamidine, the ATFs survival initially dropped to 25%, but then recovered to 183% on day 5 and 74% on day 10 post pentamidine treatment, suggesting that a significant number of parasites was not eliminated by pentamidine. A more pronounced phenotype was observed upon melarsoprol treatment where the ATFs survival dropped to 47%, but then recovered to 451% on day 5 and 1613% on day 10 post melarsoprol treatment. Altogether, these data showed that, upon drug treatment, a fraction of the ATFs population was able to evade drug treatment, survive and rise in numbers.

Overall, our results show that both slow and fast-replicating ATFs are reversible and can readapt to growing in bloodstream conditions. Importantly, the parasite tissue population is more resilient to treatment with anti-trypanosome drugs than the blood population.

## Discussion

Here, we show that the population of parasites that reside in the adipose tissue grows more slowly and synthesizes proteins at a lower rate than in the blood. Importantly, ATFs are more heterogeneous than BSFs, which could explain why the former population is more refractory to drug treatment. Together with previous observations that ATFs adapt metabolically to the tissue environment, our work shows that tissue populations are not only different from the bloodstream counterparts, but present a higher degree of heterogeneity than anticipated.

Our mathematical model analysis adds an extra layer of complexity to the published models, typically focused on antigenic variation or differentiation mechanisms[12,17,31,32]. Here we addressed the existence of differences between parasite dynamics in the blood and adipose tissue, anticipating an adipose tissue population that replicates at half the rate of the blood population, which is consistent with the temporal and magnitude trends present in the data. Besides testing for inter-compartmental variability, this modeling approach also provides estimates for other relevant parameters involved in infection, such as the migration rate between compartments and contribution to clearance by the immune response. Model 2, for instance, predicts a migration rate of about 0.11 cells/mg/day suggesting that, per milligram of organ, each day about 11% of blood parasites leave to the adipose tissue and vice-versa (0.005 per hour). This observation is in line with previously estimated migration rates between blood and organs of mice infected with *Salmonella enterica* (0.005 per hour)[33]. Similarly, the lower bound for the ratio between slender vs stumpy cells killing rates, estimated by the joint posterior of this model in the interval 6.5-24, also falls within the reported estimated range 5–7[34], although we find more uncertainty with this dataset. The remaining parameter estimates are similar between models and approximate existing values in the literature (e.g. antigen switching rate)[31,32].

Naturally, the space for modeling is infinite, but including additional complexity is a stepwise process, ideally backed by different and incremental sources of data. Having found support for one major difference between the two compartments, does not rule out the existence of other

sources of variation that may, on their own, or synergistically, contribute to further individuation of infection dynamics between the blood and adipose tissue. Future models, based on more-detailed infection time-series data (e.g. explicit antigenic variants, more time points for parasite densities, tissue-specific immune responses), combined with single-cell analysis and high-resolution microscopy, should be developed. This would allow to assess other dimensions of blood and adipose tissue variations, and integrate them in frameworks for parasite fitness in natural transmission settings[35]. Expanding the description of variant-specific traits or time-dependent phenotypes, such as gradual immune suppression, could help capture more accurately the intricacies of peak-to-peak variability that the current modeling framework does not account for. It would also be interesting to characterize other extravascular populations, by modelling other relevant and inter-linked in-host compartments, and quantify their thus far-unexplored role in chronic infections.

Slow proliferation was found first at the population-level using proteomics, measurement of protein synthesis rate and cell cycle analysis. Subsequent single-cell analysis using proliferation dyes revealed that the ATFs population was more heterogeneous than the BSFs, with about 40% of the population showing a slower proliferation rate. This population heterogeneity suggests that the mechanism of adaptation to living in the adipose tissue is more complex than cell plasticity, a process by which cells may transiently adapt to the environment by homogenously shifting the growth rate. Of note, within the limit of detection of these methods, we conclude that both ATFs and BSFs are not dormant or quiescent. Different egress from tissues, different intra-tissue anatomic localization and different sensitivity to immune response may potentially explain why this subpopulation of slow growing cells is not outgrown by fast growing parasites. Future studies should enquire the transcriptome, proteome or metabolome at the single-cell level to expand our knowledge on the biological differences between parasites that replicate at different rates. A key question will be to understand how the switch is made and what drives such adaptation.

The slow-growing ATFs reverted to a fast replicative phenotype once placed in a known favorable environment. Although, this reversion may happen within sub compartments of the adipose tissue, we find it more likely that slow growing parasites revert to fast growing parasites once they egresses from the tissue and colonize the blood. Toxoplasmosis recrudescence, for instance, has been attributed to reconversion from slow growing bradyzoites to actively replicating tachyzoites due to exposure to less stressful conditions[36,37]. Reversibility of slow growing cells from adipose tissue or potentially other organs could be an important contributor to the biological mechanism by which relapses are detected in HAT patients, including those who had no symptoms for 29 years[38].

Drug tolerance is a typical feature of slow growing cells[7,11,39]. We showed that the population of ATFs is refractory to treatment with two common drugs against *T. brucei* (pentamidine and melarsoprol). Although the reduced parasite clearance could be due to the fact that slow growing parasites are intrinsically more tolerant, we cannot exclude the possibility that the concentration of drug is lower in adipose tissue than blood. In the future, drug distribution among the different body microenvironments should be inquired, with the help of an ADME study, and the drug resistant parasites should be analyzed through single-cell RNAseq as a way to confirm their slow growing behavior. Independently from the cause(s) of drug tolerance, our study highlights a potential clinically relevant problem since the presence of slow growing cells among the ATFs population could contribute to explain treatment failure, just as it does so in other bacteria and parasitic persistent infections.

The existence of slow-growing parasites within the adipose tissue population may be an evolutionary advantageous mechanism of evading the immune response, allowing for periodical blood

repopulation. This phenotype may also cause less pathology, thus improving disease chronicity and indirectly increasing the chances of transmission. Reuter et al. have recently identified and characterized persister-like cells (skin tissue forms, STFs) in artificial human skin. These STFs are slow replicating parasites with a unique transcriptome, a downregulated metabolism, DNA synthesis and protein synthesis and the ability to revert to its actively replicating state once placed in favorable environmental conditions[40]. A burning question for future studies is whether slow growing parasites exist in multiple organs (including the brain) in vivo and if they are metabolically and functionally identical among organs. This knowledge will pave the way to a better understanding of relapses and failures of drug treatment in humans.

## Methods

### Parasite cell lines

To characterize the infection dynamics, a *T. brucei* pleomorphic stumpy reporter cell line was used (AnTat1.1 *GFP::PAD1$_{utr}$*) while for the remaining experiments, a monomorphic strain was selected (Lister 427), to avoid the mixture of the two life cycle stages (slender and stumpy forms). Procyclic forms (PCFs) were obtained from Lister 427 cultured parasites, by placing 2 million parasites in 1 mL of DTM medium with 6 mM of cis-aconitic acid at 27 °C with 5% $CO_2$ and allowing them to grow for 3 days.

### Animal infections

Animal experiments were performed according to EU regulations and approved by the Órgão Responsável pelo Bem-estar Animal (ORBEA) of Instituto de Medicina Molecular and the competent authority Direcção Geral de Alimentação e Veterinária (license number: 018889 \2016).

Mice were group-housed in filter-top cages in a Specific-Pathogen-Free barrier facility under standard laboratory conditions: 21 to 22 °C with 45 to 65% humidity and a 12 h light/12 h dark cycle. Chow and water were available ad libitum. All infections were performed in 8-13 week old wild-type male C57BL/6J mice with origin in Charles River Laboratories, France, by intraperitoneal (i.p.) injection of 2000 parasites, unless otherwise stated. Parasite viability was evaluated prior to infection under an optical microscope.

Parasites were collected from mice sacrificed by $CO_2$ narcosis. Blood parasites were collected first by heart puncture, while adipose tissue parasites collection was performed on perfused gonadal depots incubated in HMI11 (or Creek's Minimal Medium, for the proteomics assay) at 37 °C and 150 rpm for up to 70 min. Mice were manually transcardially perfused with pre-warmed heparinized saline buffer (50 mL 1x phosphate buffered saline (1xPBS) with 250 μL of 5000 I.U./mL heparine per animal). Subsequently, parasites were isolated using a DEAE sepharose™ Fast Flow column whenever necessary.

### Infection dynamics

Parasitemia was assessed daily by taking blood from the mouse-tail vein and counting the number of parasites using a Neubauer chamber. The total number of parasites present in the blood and the gonadal depot was determined by qPCR. Briefly, animals were sacrificed by $CO_2$ narcosis and blood collected by heart puncture. Blood sample was split into two. On one sample, red blood cells were lysed with Red Blood Cell Lysis Buffer (0.15 M ammonium chloride; 0.01 M potassium bicarbonate; 0.001 M EDTA disodium salt) and the remaining pellet snap frozen in liquid nitrogen. The other sample was used for parasites isolation. After perfusion, gonadal depots were collected and one was snap frozen immediately while the other was used to isolate fat parasites. Genomic DNA (gDNA) was extracted from tissues using NZY tissue gDNA isolation kit and the amount of 18 S rDNA gene of *T. brucei* present in the blood and the gonadal depots was measured by qPCR and converted into number of parasites using standard curves[3]. Isolated parasites from blood and gonadal adipose tissue were fixed with

1% paraformaldehyde, permeabilized with 0.5% Triton X-100 and stained with 10 μg of propidium iodide. Their GFP expression was then analyzed on a BD LSRFortessa™ cell analyzer with FACSDiva 6.2 and data treated using FlowJo™ 10.

In total, 106 8–12-week-old wild-type male C57BL/6J mice were used to determine infection dynamics.

### Mathematical modeling

Average parasite densities and percentage of stumpy forms obtained from both blood and adipose tissue gonadal depot during the infection dynamics characterization were used to fit the mathematical model. The ordinary differential equations for population dynamics were based on ref. [31], but in this study the number of compartments was doubled to represent parasites growing in the blood and adipose tissue, and the immune response dynamics was simplified. The model variables for each antigenic wave were the slender and stumpy forms in the two compartments, as well as the variant-specific immune response. Allowing for up to five consecutive antigenic waves during 28 days resulted in 25 inter-dependent equations. Thus, infection processes were captured with a parsimonious formulation (growth, density-dependent differentiation, antigenic variation and antigen-dependent host immune response). Further, the two in-host compartments, blood and adipose tissue, were connected through migration, assumed to occur at the same rate in either direction and for all cells. All variants were assumed symmetric, in their parameters, including the switch rate to the next wave. The majority of parameters were estimated by fitting the model to the dynamic data; with exception of a few such as, the stumpy form lifespan which was fixed at about 2 days[31] (Supplementary Model Information for detailed description on model structure and biological assumptions).

Model fitting to data was performed under a Bayesian framework, using the adaptive Monte Carlo Markov chain mcmcstat package in Matlab[41]. For models 1, 2, and 3, a total of 8, 9, and 10 parameters were estimated, respectively. Two independent Monte Carlo Markov Chains were run in each case, until convergence, which took on average 20,000 iterations. Posterior distributions were then obtained from running another 10,000 iterations post-burnin. To compare hypotheses, Deviance Information Criterion (DIC)[15], the posterior error distribution and marginal likelihood[16] were computed for each model, and the quality of fits to data and feasibility of parameters were inspected (Supplementary Data 1).

### Proteomics

Mice infected for 5 days with Lister 427 parasites were sacrificed and parasite isolation from blood and adipose tissue was performed as described above. Parasites were washed in Creeks Minimal Medium depleted from Fetal Bovine Serum and lysed with 1X NuPAGE™ LDS Sample Buffer supplemented with 100 mM DL-Dithiothreitol by boiling at 80 °C for 15 min. Parasites isolated from up to 6 mice were pooled to obtain a minimum of 0.32 million parasites. Protein samples were then separated, prepared and measured as in[42] with the exception of a 4 hours MS run. Briefly, protein samples were separated on a 4–12% NuPAGE Novex Bis-Tris precast gel (Life Technologies) for 10 min at 180 V and destained until the bands were faint. Protein reduction and alkylation was achieved using 10 mM DL-Dithiothreitol and 50 mM 2-Iodoacetamide, respectively. Trypsin digestion was performed overnight at 37 °C. Peptides were eluted and desalted using Solid Phase Extraction Disk C18 (3 M) material, reverse-phase separated using an EASYnLC 1000 HPLC system with a 25 cm capillary (75 μm inner diameter; New Objective) and self-packed with Reprosil C18-AQ 1.9 μm resin (Dr. Maisch) for chromatography. This column was coupled via a Nanospray Flex Source (ESI) to a Q Exactive Plus mass spectrometer (Thermo Fisher Scientific). Peptides were sprayed into the mass spectrometer running a 200 minute optimized gradient from 2 to 40% acetonitrile with 0.1% formic acid at a flow rate of 225 nL/min.

Measurements were performed in positive mode and with a resolution of 70,000 for full scan and resolution of 17,500 for MS/MS scan. For HCD fragmentation, the 10 most intense peaks were selected and excluded afterwards for 20 s. Protein quantification was processed in MaxQuant version 1.6.7.0[43] using standard settings and activated LFQ algorithm. The raw proteomics files were searched against the protein databases of *T. brucei* TREU927 (TriTrypDB version 33[44]), *Mus musculus* strain C57BL/6 J (UniProt), the 14 VSGs of the Lister 427 *T. brucei* strain (UniProt) and the contaminants database included in MaxQuant. Contaminants, reverse Protein Groups (PGs) and PGs only identified by a modification site or by less than 2 peptides (of which 1 needed to be unique) were removed as well as *M. musculus* identified proteins. To assign a quantification to missing values, these were imputed 1000 times using a β-distribution with equal shape parameters ($\alpha = \beta = 2$) and a PG was only considered differentially regulated if it was found up or downregulated in at least 99% of times. For each individual replicate, the obtained distribution was scaled between 0.1 and 1.5 percentile of the log2 transformed measured label-free quantitation (LFQ) intensity values. Finally, only the PGs that were quantified by LFQ intensity in at least 2 replicates of one condition (ATFs or BSFs) were considered for further analysis. Regulated PGs were determined if the relationship between the significance of the Welch's *t*-test and their fold-change was above a threshold defined by a reciprocal function (with limits *p*-value = 0.05 and fold change = 1.5).

In total, 22 9–11-week-old wild-type male C57BL/6J mice were used to define the proteomes of BSFs and ATFs.

The GO term enrichment analysis was performed in R with GO.db annotation package[45] from Bioconductor and Fisher's Exact Test (*p*-value ≤ 0.05) from stats package. *T. brucei* TREU927 GO term annotations were obtained from TriTrypDB and GO term enrichment was assessed. PGs were associated with the GO terms of the individual genes in the group and a GO term enrichment test was performed separately on upregulated and downregulated gene. Only the GO terms with at least 5 annotated PGs were considered.

The mass spectrometry proteomics data have been deposited to the ProteomeXchange Consortium via the PRIDE partner repository with the dataset identifier PXD014958.

## Protein synthesis

Differences in protein synthesis were determined through Click-iT® Homopropargylglycine (HPG) incorporation into nascent proteins. To establish the ideal concentration of HPG, cultured Lister 427 parasites were washed in methionine free Minimal Essential Medium and incubated in the same medium with 25 μM, 50 μM, or 100 μM of HPG for 30 min at 37 °C. Cultured parasites were also incubated in the same medium with 100 μg/mL of cycloheximide for 20 min at 37 °C before adding 50 μM HPG. Isolated BSFs and ATFs from mice infected for 5 days with Lister 427 parasites as well as PCFs were also incubated with 50 μM HPG under the same conditions. After incubation, all tested parasites were fixed with 1% paraformaldehyde and permeabilized with 0.5% Triton X-100. In all, 100 μL of the reaction cocktail were then added for a 30-min period at room temperature (light protected) and samples washed after with 100 μL of Reaction Rinse Buffer (Click-iT® HPG Alexa Fluor® 488 Protein Synthesis Assay Kit). All samples were stained with 1 μg/mL of 4′,6-diamidino-2-phenylindole (DAPI) and both DAPI and HPG intensities measured with BD LSRFortessa X-20 in FACSDiva 8.0. Data analyses were performed in FlowJo™ 10.

In total, 26 8–10-week-old wild-type male C57BL/6J mice were used to evaluate the differences in protein synthesis.

## Cell-cycle profile

The number of kinetoplasts and nuclei were assessed for both blood and adipose tissue populations by microscopy imaging in vivo and ex vivo. To compare the two profiles, parasites from a pool of 2 mice infected with Lister 427 parasites for 5 days were stained either by intravenous injection of 10 mg/kg of bisBenzimide H 33342 trihydrochloride (Hoechst 33342) (in vivo experiments) or by adding 1 μg/mL of Hoechst 33342 (ex vivo experiments) and observed by intravital microscopy and ex vivo microscopy, respectively. More than 600 cells were analyzed per condition.

In total, 19 8–10-week-old wild-type male C57BL/6J mice were used to assess the number of kinetoplasts and nuclei.

## DNA synthesis

Measurement of DNA synthesis was based on the incorporation of 5-ethynyl-2′-deoxyuridine (EdU) and its subsequent detection by a fluorescent azide through click chemistry. EdU was either administered intravenously (200 mg/kg) to mice infected for 5 days with Lister 427 parasites (in vivo protocol), or added (100 μM) to parasites isolated from mice infected for the same period and cultured in HMI11 at 37 °C (ex vivo protocol). In both conditions, parasites were incubated with EdU for 30 min. The remaining protocol was conducted mostly according to manufacturer's instructions with some minor changes (ThermoFisher Scientific). Briefly, parasites were washed with cold 1xTDB, fixed with 1% paraformaldehyde for 10 min followed by quenching with 0.125 M glycine for 5 min. Parasites were adhered to silanized coverslips for 30 min, permeabilized with 0.5% Triton X-100 for 20 min, washed twice with 3%BSA in 1xPBS and fluorescently labeled with Alexa Fluor® 488 fluorescent azyde by exposure to 300 μL of Click-iT® reaction cocktail for an extra 30 min, protected from light. Finally, cells were washed twice with 3%BSA in 1xPBS, stained with 1 μg/mL of Hoechst 33342 for 30 min and the percentage of EdU positive cells was assessed by fluorescence microscopy. All the reported steps were performed at room temperature. More than 300 cells were analyzed per condition.

In total, 19 8–10-week-old wild-type male C57BL/6J mice were used to measure the DNA synthesis.

## Cell proliferation

10 million Lister 427 cultured parasites were labeled with 2 μM of CellTrace™ Violet in 10 mL 1xPBS for 20 min at 37 °C, protected from light. Remnants of free dye were removed by adding 5 mL of HMI11 to the cells for 5 min at 37 °C. Pelleted parasites were then resuspended in HMI11. A fraction of these parasites was fixed with 1% paraformaldehyde for 10 min followed by quenching with 0.125 M glycine for 5 min and pellet resuspended in 3%BSA/0.05% TX-100 in 1xPBS and observed by microscopy to determine basal die incorporation. Also, 1 million labeled parasites were immediately intraperitoneally injected in a mouse. As a control, mice were also infected with non-labeled parasites. Mice parasitemia was followed daily by taking blood from the mouse-tail vein and counting the number of parasites using a Neubauer chamber. Two days post-infection, adipose tissue parasite density and the percentage of motile parasites were also determined for both conditions by intravital microscopy analysis. More than 150 cells were analyzed per condition. In addition, the Mean Fluorescence Intensity (MFI) levels of CTV within blood and adipose tissue parasites were assessed either by intravital microscopy or by ex vivo microscopy analyses. More than 180 cells were analyzed per condition. The doubling time was estimated by dividing the infection time by the number of divisions. The number of divisions of each population (x) was obtained from the following equation: $y = C0e^{-0.693x}$, where y is the determined average mean fluorescence intensity of the analyzed population and C0 is the initial mean fluorescence intensity of the population used to infect mice. In total, 26 9–11-week-old wild-type male C57BL/6J mice were used to estimate the doubling times.

Labeled parasites were also used to determine the reversion of the slow growing phenotype. For that, mice were infected with 0.1 million Lister 427 parasites and three days post infection ATFs were isolated and sorted for CTV positive and negative populations. All sorted parasites were resuspended in HMI11 and either intraperitoneally

injected in a mouse or incubated at 37 °C with 5% $CO_2$ for a maximum period of 12 days. Parasite growth was followed daily and the number of parasites counted with a Neubauer chamber. For several technical reasons (e.g. sorting, dilution, growth conditions), in 2 out of 4 in vitro performed experiments no healthy growth conditions were observed for up to 1 week. In total, 40 10–13-week-old wild-type male C57BL/6J mice were used to define the reversion process.

2 million Lister 427-cultured parasites were labeled with 1 µM of PKH26 Red fluorescent cell linker in Diluent C for 5 min at room temperature with periodic shaking (light protected). Remnants of free dye were removed by adding 10 mL of HMI11 to the cells for 1 minute at 37 °C. Pelleted parasites were then washed in HMI11 three times and resuspended in HMI11. To a fraction of these parasites 0.04% paraformaldehyde was added and the parasites were observed by microscopy to determine die incorporation. Another fraction was used to determine progressive PKH26 signal decay. For that, three cultures of initially 0.2 million parasites per milliliter of HMI11 were followed for a 50 h period with periodic flow cytometry analysis of the PKH26 fluorescence intensity (0, 5, 10, 20, 25, 29, 35, 44, and 50 h). Finally, 0.02 million parasites were immediately intraperitoneally injected in each mouse. Three and four days post-infection, the Mean Fluorescence Intensity (MFI) levels of PKH26 within blood and adipose tissue isolated parasites were assessed with BD LSRFortessa X-20 in FACSDiva 8.0. Data analyses were performed in FlowJoTM 10. The Proliferation Platform was used to calculate the doubling times of the PKH26 labeled cultured parasites. In total, 12 12–13-week-old wild-type male C57BL/6J mice were used to exclude the presence of non-dividing ATFs.

## Mice drug treatment
Mice infected for 4 days with Lister 427 were treated with a single intraperitoneal dose of either 20 mg/kg of pentamidine or 10 mg/kg of melarsoprol. For the 3 days after, mice parasitemia was followed daily by taking blood from the mouse-tail vein and counting the number of parasites using a Neubauer chamber. 3, 5, and 10 days post treatment the percentage of ATFs survival was determined by intravital microscopy analysis.

In total, 32 9–10-week-old wild-type male C57BL/6J mice were used to study drug evasion by the parasites.

## Microscopy
Imaging of the PKH26 signal (551 nm) was performed on the confocal point-scanning microscope with Airyscan Zeiss LSM 880 using a 63x objective lens (Plan-Apochromat, NA 1.40, oil immersion, Zeiss). DPSS 561-20 Laser Unit and Diode 405-30 were used for visualizing PKH26 and for bright field, respectively. All the remaining imaging was performed on a 3i Marianas spinning disc confocal microscope (Intelligent Imaging Innovations) using a ×63 objective lens (Plan-Apochromat, NA 1.40, oil immersion, Zeiss). Laser stacks 405 (405 nm) and 488 (488 nm) were used for visualizing kinetoplasts and nuclei (Hoechst) or CTV, and Fluorescein Isothiocyanate-Dextran (FITC-Dextran), respectively. To avoid photodamage, an average maximum laser power of 2 mW, a gain value of 1 and an exposure of 100 milliseconds were selected. A time lapse of 5 s was acquired, with images obtained every 10 milliseconds. In addition, bright field imaging was performed for visualization of infected parasites in the blood monolayer and BSFs and ATFs ex vivo. Background correction was performed with internal controls (i.e. regions without parasites in CTV positive samples for CTV experiments and cytoplasmic regions of parasites for EdU experiments) and exogenous controls (i.e. parasites in CTV- or EdU-negative samples). Segmentation (CTV experiments) was based on the bright field (blood) or 488 nm (adipose tissue and blood) detections, and mean fluorescence intensity (MFI) was calculated for each parasite based on the CTV signal (405 nm), Hoechst 33342 signal (405 nm), or Click-iT™-EdU signal (488 nm). Parasites were imaged across the full

available sample using a "snake by rows" observation and acquisition approach covering the entire tissue/dish. For all tested conditions, a minimum of 50 parasites per sample was imaged in a minimum of 25 different fields of view. Acquired images were then analyzed in Fiji 2.9.0/1.53t[46] and Ilastik (https://www.ilastik.org).

## Sample preparation for microscopy
The representative image of PKH26 labeled parasite was performed on cultured parasites labeled with 1 µM PKH26. Labeled cells were washed twice with 500 µL of 1X PBS, pelleted by centrifugation for 5 min at 2000 rpm and resuspended in 50 µL of 1X PBS with 0,004% paraformaldehyde (PFA). In all, 1 µL was deposited in a slide with a coverslip.

Prior to intravital microscopy, mice were anaesthetized with 120 mg/Kg ketamine and 16 mg/Kg xylazine injected intravenously. FITC-Dextran 70 kDa was also injected intravenously immediately preceding imaging to distinguish between blood vessels and tissue parenchyma. The gonadal adipose tissue was exposed through a small incision of 20-40 mm made on the lower abdominal region of the mouse and a temporal glass window (Merck rectangular coverglass, 100 mm × 50 mm) implanted for imaging. The mouse was then placed on the microscope stage, and the infected blood vessels from the adipose tissue and the adipose tissue per se, were imaged to assess a sample of the BSFs and ATFs population, respectively. This protocol was applied to both gonads. BSFs population was also assessed by imaging blood monolayers obtained from 2 µL of collected blood, diluted in 200 µL of 1xPBS and placed on a petri dish. Non-motile parasites were excluded from the analysis.

For ex vivo microscopy, isolated ATFs and BSFs were concentrated into 200 µl by centrifugation for 2 min at 10,000×$g$. The 200 µl were then transferred to a glass bottom dish (Matek Life Sciences, 60 mm, No. 1.5 thickness) and imaged.

## Statistical analysis
Statistical analyses were performed in the free software R: http://www.r-project.org version 4.1.2. At least three independent experiments were considered in each case and statistical significance was set to $\alpha = 0.05$ level. Data were analyzed after logarithm transformation. Statistically significant equal doubling time of CTV-negative and CTV-positive ATFs was determined by a Wilcoxon rank sum test. The differential percentage of stumpy forms between BSFs and ATFs and the reduced protein synthesis in ATFs versus BSFs were determined by Wilcoxon signed rank tests. To define the differential estimated doubling time of BSFs and ATFs we fitted a linear mixed-effects model, using the nlme package in R, considering mice as random factors. The bias in EdU incorporation for BSFs relative to ATFs was also determined by fitting a generalized linear mixed-effects model, using the lmer package in R, considering mice and organs as random factors.

Company names and catalog numbers of commercial reagents are provided in Supplementary Data 3.

## Reporting summary
Further information on research design is available in the Nature Portfolio Reporting Summary linked to this article.

# Data availability
The number of independent replicates was pre-estimated by a power analysis performed on G*Power 3.1 software. The used data sets were anterior to this publication (2 different groups of infected mice). TriTrypDB version 33 and UniProt protein databases were used in the proteomic analysis. The mass spectrometry proteomics data generated in this study have been deposited to the ProteomeXchange Consortium via the PRIDE partner repository with the dataset identifier PXD014958. The proteomic analysis generated in this study is provided in the Supplementary Data 1. Source Data are provided with this paper. Figures 1 and 2 are linked to Supplementary Table 1. Figure 2 is

linked to Supplementary Information, Supplementary Data 1 and Supplementary Tables 2 and 3. Figure 3 is linked to Supplementary Data 2 and Supplementary Table 4. Company names and catalog numbers of commercial reagents are provided in Supplementary Data 3. Source data are provided with this paper.

## Code availability

All technical aspects related to modeling and model-fitting to data are fully detailed in the Supplementary Information. Codes are available in the Supplementary Data 1.

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

## Acknowledgements

The authors thank Keith Matthews (University of Edinburgh) for providing AnTat1.1E clone, Christian Janzen (University of Wurzburg) for the *GFP::PAD1_{utr}* cell line, Pere Perez-Simarro for providing melarsoprol. Ruy M. Ribeiro for useful discussions about the statistical analysis and Helena Pinheiro for drawing the diagram of the mathematical infection model. The authors would like to thank all members of the Figueiredo and Prudêncio laboratories for helpful discussions and reagents. We also acknowledge the Rodents, Bioimaging and Flow Cytometry Units of the Instituto de Medicina Molecular. The project leading to this application has received funding from the European Research Council (ERC) under the European Union's Horizon 2020 research and innovation programme (grant agreement No 771714). This work was also supported by IC&DT Programa de Actividades Conjuntas – ref. 016417 "Oneida", PTDC/BIMMET/4471/2014 and PTDC/CVT-CVT/29161/2017 (FCT). SFRH/BPD/89833/2012 and DL 57/2016/CP1451/CT0019 (FCT) to S.T, LT000047/2019-L (HFSP) and ALTF 1048-2016 (EMBO) to M.D.N, MSCA ITN Cell2Cell fellowship to L.LE and CEECIND/03322/2018 (FCT) to L.M.F. FBr is supported by the Agence National de Recherche (ANR, grant number ANR19-CE15-0004-01: ADIPOTRYP), the Fondation pour le Recherche Médicale (FRM, grant n°EQU201903007845: "Equipe FRM") and the Laboratoire d'Excellence (grant number ANR-11-LABX-0024:ParaFrap).

## Author contributions

S.T., M.D.N., J.F., E.G., and L.M.F. designed research. S.T., M.D.N., M.C.Q., T.B.R., F.Be, L.L.E., M.V.N, F.Bu, and E.G. performed experiments. S.T., M.D.N., M.C.S., M.D., F.Br, E.G., and L.M.F. analyzed data. S.T., F.Br, E.G., and L.M.F. wrote the paper.

## Competing interests

The authors declare no competing interests.
