## [Peer Review File · Nature Communications]

Reviewer comments, first round review

Reviewer #1 (Remarks to the Author):

The authors were asked to be careful on the semantics around the cell population they were looking at and they have removed the term "persister" and focus instead on the fact that the ATF parasites grow more slowly than their bloodstream form counterparts, which is a more accurate depiction.

The biochemical changes between BSF and ATF are clearly shown.

Irrespective of what one calls the parasites the authors have shown that the ATF parasites are somewhat less susceptible to treatment with two trypanocidal drugs than are BSF organisms. They have added a note of caution that this could be due to the biochemical changes between forms, but might also involve other factors including differences in how these drugs reaches parasites in blood vs adipose tissue.

As such they have addressed the points I had requested at their second submission to Nature Micro.

Reviewer #3 (Remarks to the Author):

I am stepping in for a reviewer at a late stage. From reviewing the the manuscript, referees' comments and response to the referees, I think the manuscript has been substantially changed to tone down the initial claims made. I find the manuscript to be sound and covers a lot of work. However, the findings are less noteworthy than if the authors had indeed discovered drug resistant persisters.

Reviewer #4 (Remarks to the Author):

I would like to thank the authors for responding to my previous comments so carefully and extensively. I believe the consideration of the extra model provides good evidence regarding the robustness of the (modelling) results and conclusions. It is clear that based on the available data, making precise estimates of individual parameter values is not possible, and concentrating instead on the overall message (growth rate differences between tissues) will be sufficient from my point of view. I therefore also do not think it necessary to include the new model in the main section; although it would definitely be worth it trying to run it for more iterations / convergence, which could then be added as supplementary info.

REVIEWERS' COMMENTS

We thank the three reviewers for the final positive opinions.

Reviewer #1 (Remarks to the Author):

The authors were asked to be careful on the semantics around the cell population they were looking at and they have removed the term "persister" and focus instead on the fact that the ATF parasites grow more slowly than their bloodstream form counterparts, which is a more accurate depiction.

The biochemical changes between BSF and ATF are clearly shown.

Irrespective of what one calls the parasites the authors have shown that the ATF parasites are somewhat less susceptible to treatment with two trypanocidal drugs than are BSF organisms. They have added a note of caution that this could be due to the biochemical changes between forms, but might also involve other factors including differences in how these drugs reaches parasites in blood vs adipose tissue.

As such they have addressed the points I had requested at their second submission to Nature Micro.

Reviewer #3 (Remarks to the Author):

I am stepping in for a reviewer at a late stage. From reviewing the the manuscript, referees' comments and response to the referees, I think the manuscript has been substantially changed to tone down the initial claims made. I find the manuscript to be sound and covers a lot of work. However, the findings are less noteworthy than if the authors had indeed discovered drug resistant persisters.

Reviewer #4 (Remarks to the Author):

I would like to thank the authors for responding to my previous comments so carefully and extensively. I believe the consideration of the extra model provides good evidence regarding the robustness of the (modelling) results and conclusions. It is clear that based on the available data, making precise estimates of individual parameter values is not possible, and concentrating instead on the overall message (growth rate differences between tissues) will be sufficient from my point of view. I therefore also do not think it necessary to include the new model in the main section; although it would definitely be worth it trying to run it for more iterations / convergence, which could then be added as supplementary info.